# Connexins evolved after early chordates lost innexin diversity

Georg Welzel*, Stefan Schuster*

Department of Animal Physiology, University of Bayreuth, Bayreuth, Germany

**Abstract** Gap junction channels are formed by two unrelated protein families. Non-chordates use the primordial innexins, while chordates use connexins that superseded the gap junction function of innexins. Chordates retained innexin-homologs, but N-glycosylation prevents them from forming gap junctions. It is puzzling why chordates seem to exclusively use the new gap junction protein and why no chordates should exist that use non-glycosylated innexins to form gap junctions. Here, we identified glycosylation sites of 2388 innexins from 174 non-chordate and 276 chordate species. Among all chordates, we found not a single innexin without glycosylation sites. Surprisingly, the glycosylation motif is also widespread among non-chordate innexins indicating that glycosylated innexins are not a novelty of chordates. In addition, we discovered a loss of innexin diversity during early chordate evolution. Most importantly, lancelets, which lack connexins, exclusively possess only one highly conserved innexin with one glycosylation site. A bottleneck effect might thus explain why connexins have become the only protein used to form chordate gap junctions.

**\*For correspondence:**
georg.welzel@uni-bayreuth.de
(GW);
stefan.schuster@uni-bayreuth.de (SS)

**Competing interest:** The authors declare that no competing interests exist.

## Introduction

Animals from humans to comb jellies use gap junction channels to couple adjacent cells and thus enable direct intercellular communication. Interestingly, gap junction channels are formed by two unrelated integral membrane proteins: innexins and connexins. The innexins are the primordial gap junction proteins that have been identified in all eumetazoans except sponges, placozoa, and echinoderms (*Slivko-Koltchik et al., 2019*). The connexins arose de novo during the early chordate evolution and constitute the gap junction channels of all living chordates except lancelets (*Abascal and Zardoya, 2013*; *Mikalsen et al., 2021*; *Slivko-Koltchik et al., 2019*). Both types of intercellular gap junction channels are formed by the docking of two hemichannels of two neighboring cells. Each hemichannel is composed of six connexins or eight innexins, respectively (*Skerrett and Williams, 2017*). As connexins and innexins are encoded by large gene families (*Mikalsen et al., 2021*; *Yen and Saier, 2007*), different connexin- and innexin-based gap junction channels can be formed to fulfill unique functions. Moreover, a diversity of innexins (or connexins) co-expressed in the same cell can dramatically increase the functional diversity in both types of gap junction channels. This is because each hemichannel can freely combine the N different co-expressed innexins or connexins so that theoretically up to $N^6$ (connexins) or $N^8$ (innexins) heteromeric hemichannels would be possible.

Despite the lack of sequence homology (*Alexopoulos et al., 2004*), the topology (*Maeda et al., 2009*; *Michalski et al., 2020*; *Oshima et al., 2016*; *Figure 1A and B*) and function (*Pereda and Macagno, 2017*; *Skerrett and Williams, 2017*) of connexin- and innexin-based gap junction channels are remarkably similar. Nevertheless, it is thought that chordates have completely replaced the innexin-based gap junctions with the novel connexin-based gap junctions. Vertebrates still express innexin-homologs (*Baranova et al., 2004*; *Panchin et al., 2000*), called pannexins, but it is supposed that these stopped forming gap junctions and since then only function as non-junctional membrane channels (*Dahl and Muller, 2014*; *Esseltine and Laird, 2016*; *Sosinsky et al., 2011*). This hypothesis is based on the discovery that the three pannexins of humans and mice are glycoproteins. Each of the pannexins contains an identified consensus motif (Asn-X-Ser/Thr) for asparagine (N)-linked

glycosylation within either the first or the second extracellular loop (EL; *Penuela et al., 2007*; *Penuela et al., 2014a*; *Ruan et al., 2020*; *Sanchez-Pupo et al., 2018*). This enables the posttranslational attachment of sugar moieties at the asparagine residue within the consensus sequence which hinders two pannexin channels of adjacent cells to form intercellular channels (*Ruan et al., 2020*; *Figure 1C*). Based on these findings, it has been assumed that each vertebrate pannexin is equipped with an N-linked glycosylation site (NGS) and thus lost its gap junction function (*Sosinsky et al., 2011*). However, it remains unclear whether really all vertebrate pannexins are glycosylated and thus presumably only function as single membrane channels. It is also unknown whether glycosylation is indeed a novel modification gained by chordates to prevent their innexin-homologs from forming gap junctions. Specifically, previous studies have shown that at least two non-chordate species, *Aedes aegypti* (*Calkins et al., 2015*) and *Caenorhabditis elegans* (*Kaji et al., 2007*), possess an innexin protein with an extracellular NGS that is glycosylated. Additionally, putative NGS have also been described in a terrestrial slug (*Limax valentianus*) (*Sadamoto et al., 2021*). These findings raise the intriguing possibility that N-glycosylation might actually be rather common in both chordate and non-chordate innexins, and that N-glycosylation might have played an important role in the evolution of gap junction proteins.

Since the experimental identification of N-glycosylated proteins is technically demanding, time-consuming, and expensive, accurate computational methods are commonly used to identify NGSs in primary amino acid sequences (*Gupta and Brunak, 2002*; *Pitti et al., 2019*). In this study, we used the wealth of data that is now available in several public protein, genome, and transcriptome databases to analyze the occurrence of NGSs in non-chordate and chordate innexins in silico. Based on our findings, we suggest a new evolutionary scenario in which a loss in innexin diversity could explain why connexins arose de novo during the early chordate evolution and why connexins have completely replaced the innexins that so successfully serve diverse functions in the nervous systems of invertebrates.

## Results and discussion

We first screened for innexin proteins across multiple non-chordate taxa by using innexin proteins as sequence queries in BLAST searches. Only hits that fulfilled defined criteria were included in our study (for more details, see Materials and methods). In total, we collected the amino acid sequences of 1521 non-chordate innexins from 174 species across nine higher-level taxonomic groups (ctenophores, cnidarians, molluscs, annelids, platyhelminthes, nematodes, arthropods, xenacoelomorphs, and echinoderms). We subsequently searched in each of the sequences for the consensus motif for N-glycosylation (Asn-X-Ser/Thr). As the extracellular glycosylation of pannexins hinders the gap junction formation (*Ruan et al., 2020*), we only included NGSs that are located in the ELs of the innexins in our study. Surprisingly, we found that innexins with extracellular NGSs are widespread among the examined non-chordate phyla, comprising 67% of the innexins in the ctenophores (*Figure 1D and E*, *Figure 1—source data 1*). The position of the NGSs within the ELs as well as the residues around the N-glycosylation consensus motifs are not conserved between the phyla (*Figure 1F*). Within the single phyla, we found some innexin orthologs that have highly conserved NGSs and ELs (*Figure 1—figure supplement 1*). However, we did not find any extracellular NGS that was conserved in all species within a phylum. This finding is presumably based on the phylum-specific diversification of innexins. As shown in previous studies (*Abascal and Zardoya, 2013*; *Hasegawa and Turnbull, 2014*; *Moroz et al., 2014*), and demonstrated in *Figure 1D*, innexins originated early in metazoan evolution and have undergone diversification within the different non-chordate phyla. Thus, innexins with extracellular NGSs evolved independently numerous times within the single phyla.

Another point of interest is that connexins were clearly absent in all species. But there is one apparent exception, the red sea urchin (*Mesocentrotus franciscanus*). Here, a connexin (GenBank accession number GHJZ01010392.1) appears evident in the transcriptome (*Wong et al., 2019*) but is absent in the genomic data (*Sergiev et al., 2016*). Moreover, the putative connexin is clearly absent in any of the 41 available transcriptomes of other echinoderm species, including close relatives (*Strongylocentrotus purpuratus* and *Hemicentrotus pulcherrimus*). If it could be confirmed, it would be an interesting case of an aberrant and remarkably singular occurrence of a connexin.

Notably, our survey also solved an old paradox in the ambulacrarians that form gap junctions (as identified by electron microscopy and voltage-clamp experiments) (*Andreuccetti et al., 1987*;

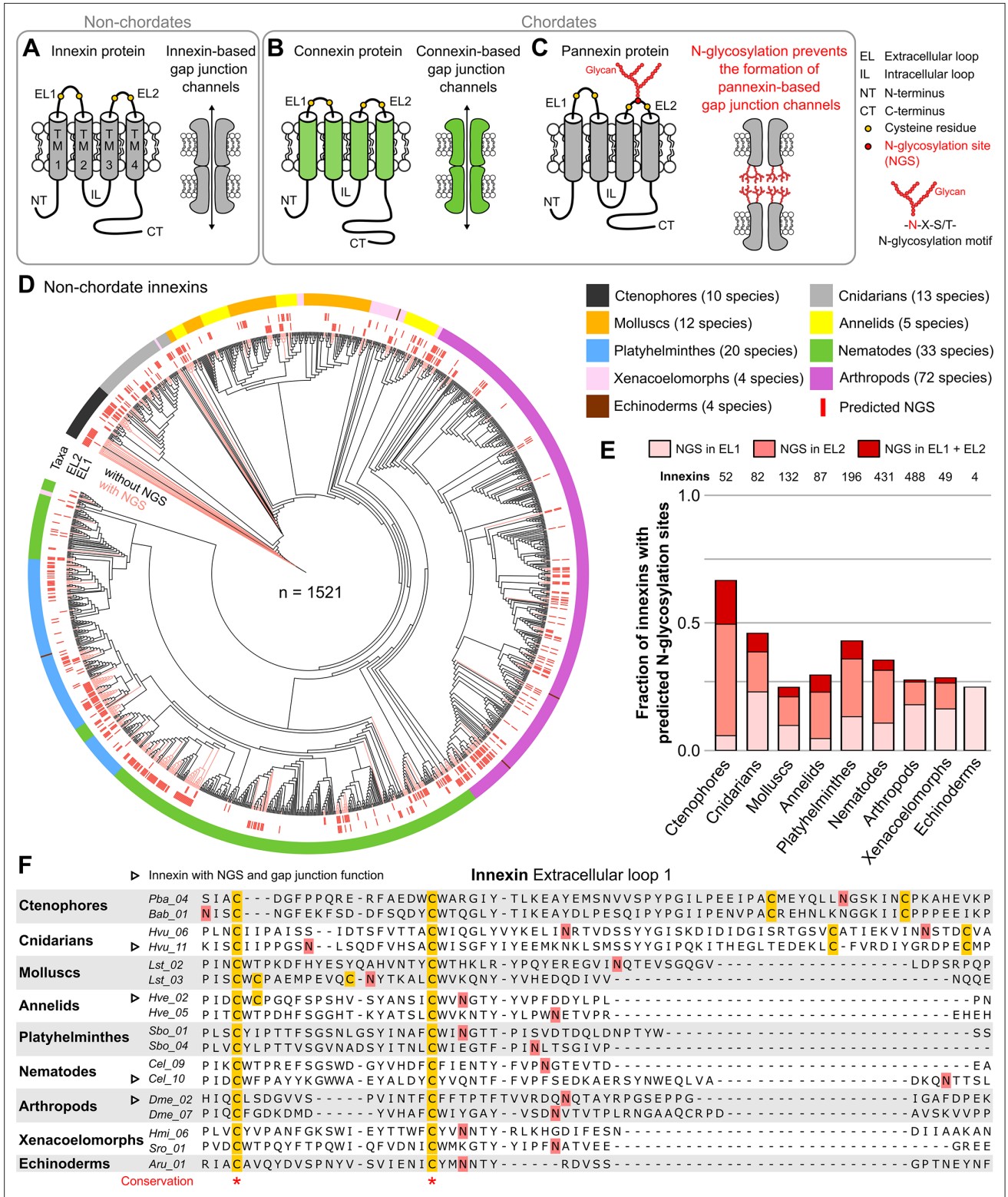

**Figure 1.** Innexins with N-linked glycosylation sites are widespread among non-chordate animals. (**A**) Innexins and (**B**) connexins are integral membrane proteins that share a common membrane topology with four hydrophobic transmembrane domains (TM) linked by one intracellular (IL) and two extracellular loops (ELs). Connexin and innexin proteins can assemble to form a hemichannel. The hemichannels of two neighboring cells can form intercellular gap junction channels that are stabilized by disulfide bonds between cysteine residues in their ELs (***Dahl et al., 1991***) (yellow circles). (**C**) The innexin homologs of vertebrates, called pannexins, are glycoproteins that contain N-linked glycosylation consensus sites (NGS) within their ELs. The attachment of glycans to NGSs block the disulfide bond formation and hence prevents the proper docking of the hemichannels and the formation

*Figure 1 continued on next page*

*Figure 1 continued*

of pannexin-based gap junctions (*Ruan et al., 2020*). (**D**) Maximum likelihood phylogeny of innexin proteins from 174 non-chordate species (n=1521 innexins). Black and red branches represent innexins without and with NGSs in the ELs, respectively. The red bars in the inner circle indicate whether EL1 or EL2 contains the NGS. The colors of the outer circle represent the phylum classification of the innexins. (**E**) Fraction of innexins with NGSs in their ELs in the taxonomic groups shown in (**D**). (**F**) Representative multiple sequence alignment of the EL1 of innexins. The alignment contains selected innexin sequences of representative species of each taxonomic group. All identified non-chordate innexins as well as the complete alignment are provided as a supplementary files (*Figure 1—source data 2* and *Figure 1—source data 4*). Conserved residues are highlighted (*, absolutely conserved). The cysteine residues (yellow) and the NGSs (red) in each EL1 are colored (note that some innexins have more than two cysteines). The identified NGSs in EL1 and EL2 of the innexins of all other non-chordate species are shown in *Figure 1—source data 1*. Arrowheads mark innexins with NGSs that have been shown to form functional gap junction channels (*Figure 1—source data 3*).

The online version of this article includes the following source data and figure supplement(s) for figure 1:

**Source data 1.** A list of the identified N-glycosylation sites (NGSs) in the extracellular loops of innexins in non-chordate species.

**Source data 2.** Multiple sequence alignment of all non-chordate innexins.

**Source data 3.** Extracellular N-glycosylation sites (NGSs) in non-chordate innexins with confirmed gap junction function.

**Source data 4.** Protein sequences of the 1521 non-chordate innexins identified in this study.

**Figure supplement 1.** The N-linked glycosylation sites (NGSs) in the extracellular loops (ELs) of innexins are conserved in some orthologs.

*Slivko-Koltchik et al., 2019*; *Yazaki et al., 1999*) but were assumed to lack both innexins and connexins (*Abascal and Zardoya, 2013*; *Burke et al., 2006*; *Slivko-Koltchik et al., 2019*), causing speculations on the nature of their junctions. Our identification of innexins in ambulacrarian transcriptomes (*Figure 1D–F* and *Figure 1—source data 1*) provides a simple solution of this puzzle.

The wide occurrence of innexins with NGSs in all non-chordate phyla (*Figure 1D and E*) as well as the experimentally confirmed NGSs (*Calkins et al., 2015*; *Kaji et al., 2007*) strongly suggest that a large fraction of non-chordate innexins are glycoproteins. These glycosylated innexin channels might then also not be able to form gap junction channels but rather function as non-junctional channels. It is interesting to note that seven of the innexins with identified NGSs have previously been shown to be parts of functional gap junction channels (*Figure 1F*, *Figure 1—source data 3*). The NGS were not conserved across these seven innexins (*Figure 1F*, *Figure 1—source data 3*) and we found no of the indications (*Petrescu, 2003*) that would suggest that these NGS might not be glycosylated.

Our findings demonstrate that non-chordate animals possess a vast diversity of innexins, with and without NGSs, and thus presumably function either as non-junctional membrane channels or as intercellular gap junction channels. This finding is in sharp contrast to the situation in chordates, where all innexins are assumed to be glycosylated and unable to form gap junction channels (*Dahl and Muller, 2014*; *Esseltine and Laird, 2016*; *Sosinsky et al., 2011*). But is there really not a single chordate species that uses pannexin-based gap junctions? Up to now, extracellular NGSs were only identified in human (*Ruan et al., 2020*), mouse (*Penuela et al., 2007*), rat (*Boassa et al., 2007*), and zebrafish (*Kurtenbach et al., 2013*; *Prochnow et al., 2009*) pannexins. To clarify the prevalence of extracellular NGSs in chordates, we again used public databases to screen for innexin proteins across multiple chordate taxa. In total, we collected the amino acid sequences of 867 chordate innexins from 276 species across nine higher-level taxonomic groups (lancelets, tunicates, lampreys, cartilaginous fish, bony fish, amphibians, reptiles, birds, and mammals) and then searched, as described above, in the ELs of each of the sequences for the consensus motif for N-glycosylation (Asn-X-Ser/Thr).

Our results clearly show that each single innexin in every chordate species has at least one NGS in its ELs (*Figure 2A and B*, *Figure 2—source data 1*). The NGSs are highly conserved in lancelets, lampreys and across all vertebrate taxa. NGS were, however, not conserved across the tunicates (*Figure 2C–E*), which may be consistent with their unusually high amino acid evolutionary rates (*Berná and Alvarez-Valin, 2014*). Across the different vertebrate taxa, the sequences of the ELs as well as the positions of the glycosylation motifs are highly conserved (*Figure 2F–H*). Among the three pannexins, conservation is particularly high in the ELs of Pannexi 2. Interestingly, conservation is still seen even after the whole-genome duplication in the common ancestor of teleost fishes (*Glasauer and Neuhauss, 2014*), an event that generally provides a source of genetic raw material for evolutionary innovation and functional divergence. Still, each single species retained their pannexins with NGSs (*Figure 2—source data 1*). This is remarkable because a single mutation in the N-glycosylation motif might be sufficient to recover the ability of pannexins to form gap junction channels (*Ruan et al.,*

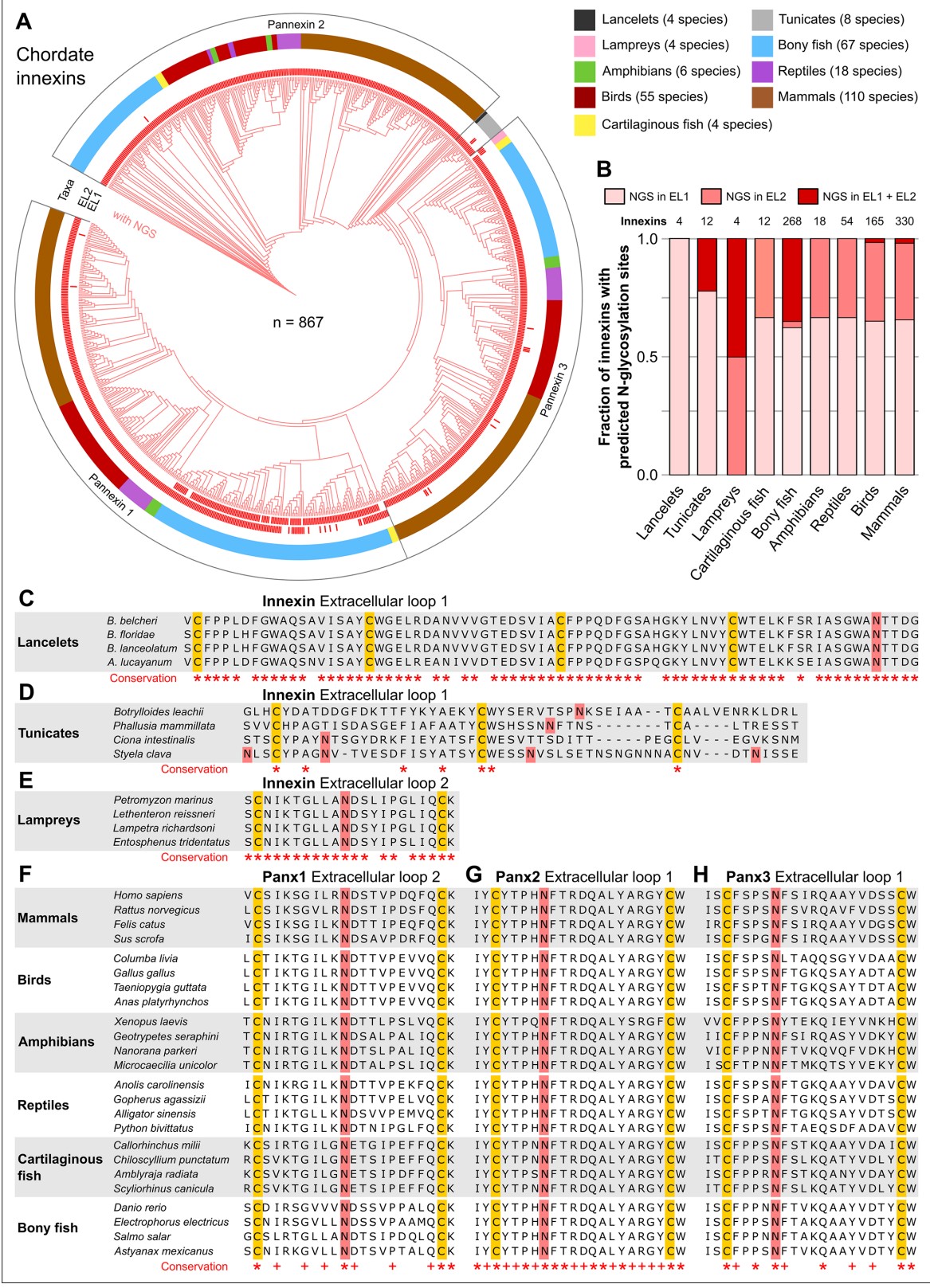

**Figure 2.** N-linked glycosylation of innexins is highly conserved in chordate animals. (**A**) Maximum likelihood phylogeny of innexin proteins from 276 non-chordate species (n=867 innexins). Red branches represent innexins with N-glycosylation consensus sites (NGS) in the extracellular loops (ELs), respectively. Note that each innexin in every chordate species has at least one NGS. The red bars in the inner circle indicate whether extracellular loop 1 or 2 (EL1 and EL2) contains the NGS. The colors of the outer circle represent the phylum classification of the innexins. (**B**) Fraction of innexins with

*Figure 2 continued on next page*

*Figure 2 continued*

NGSs in their ELs in the taxonomic groups shown in (**A**). Multiple sequence alignments of the EL1 of innexins in (**C**) lancelets and (**D**) tunicates. Note that lancelets and tunicates have more than two cysteine residues in EL1, like cnidarians and ctenophores. (**E**) Multiple sequence alignments of the EL2 of innexins in lampreys. Representative multiple sequence alignments of the highly conserved EL2 of (**F**) Pannexin 1 and the EL1 of (**G**) Pannexin 2 and (**H**) Pannexin 3 in vertebrates. Residues that are conserved in all innexins in each alignment are highlighted (*, absolutely conserved; +, physicochemical properties are conserved). Note that the pannexin alignments only contain the sequences of some representative species of each taxonomic group. All identified chordate innexins as well as the complete alignment are provided as supplementary files (*Figure 2—source data 2* and *Figure 2—source data 3*). The cysteine residues (yellow) and the NGSs (red) in each EL are colored. The identified NGSs in EL1 and EL2 of the innexins of all other chordate species are shown in *Figure 2—source data 1*.

The online version of this article includes the following source data for figure 2:

**Source data 1.** A list of the identified N-glycosylation sites (NGS) in the extracellular loops of innexins in chordate species.

**Source data 2.** Multiple sequence alignment of all chordate innexins.

**Source data 3.** Protein sequences of the 867 chordate innexins identified in this study.

---

*2020*). The surprisingly high conservation of the location and the surrounding sequences of NGSs strongly suggests that at least the vertebrate innexins serve essential roles, with correspondingly high stabilizing selective pressures (*Abascal and Zardoya, 2013*).

In summary, we show that N-glycosylation is present in both non-chordate and chordate species. Already simple organisms at the beginning of the metazoan evolution attached sugar moieties to some of their innexins to presumably prevent them from forming gap junction channels. In consequence, the vertebrate pannexins did not diverge and change their function driven by the appearance of the connexins but rather originate from an innexin already equipped with NGS. This would be consistent with findings that single membrane channels formed by pannexins and innexins have the same physiological functions and are similar in their biophysical and pharmacological properties (*Dahl and Muller, 2014*).

If it is typical for invertebrates to use a great diversity of glycosylated and non-glycosylated innexins and to even form gap junctions from both (*Figure 1—source data 3*), then the situation in the vertebrates becomes even more puzzling: Why do all vertebrates exclusively retain glycosylated innexins? Why do they not form gap junctions from them (*Ruan et al., 2020*) and instead evolved and exclusively use the new connexins for functions that could equally be fulfilled by an innexin? We suggest that looking at the early chordate evolution may solve this puzzle. Chordates are comprised of three subphyla: the lancelets, the tunicates, and the vertebrates. The lancelets represent the most basal chordate lineage that diverged before the split between tunicates and vertebrates (*Putnam et al., 2008*). The vertebrates split into the jawless fish (lampreys), the most ancient vertebrate group (*Smith et al., 2018*), and the jawed vertebrates (*Figure 3A*). As our previous analysis revealed, the lancelets and the lampreys as well as most of the tunicates have only one innexin. In the jawed vertebrates, we found three innexins (called pannexins) in each single species. This is expected from the two whole-genome duplications at the early vertebrate lineage leading first to Pannexin-2 and afterward to Pannexin-1 and Pannexin-3 (*Abascal and Zardoya, 2013*; *Fushiki et al., 2010*). Only teleost fishes have a fourth pannexin generated by a whole-genome duplication event during the teleost evolution (*Bond et al., 2012*; *Glasauer and Neuhauss, 2014*). The limited genetic diversity is thus in strong contrast to the rich innexin diversity within the non-chordate phyla (*Figures 1D and 3A*; *Abascal and Zardoya, 2013*; *Hasegawa and Turnbull, 2014*; *Moroz et al., 2014*). Here, the large gene families and the co-expression of several innexins allow impressive numbers of combinations of innexins to form heteromeric and heterotypic gap junction channels leading thus to a rich diversity of functionally unique channels (*Hall, 2017*). With the loss of innexin diversity, this aspect of functional complexity is clearly missing in chordates. Furthermore, we found that each of the innexins of lancelets, tunicates, and lampreys has an extracellular NGS (*Figures 2A–E , and 3C*). Specifically, in lancelets the only innexin available contains an NGS in its EL1. Hence, lancelets, not only lost the diversity of innexins but retained one that might not even be capable of forming gap junctions. Our work and earlier studies *Mikalsen et al., 2021*; *Slivko-Koltchik et al., 2019* found no connexin-like sequences in lancelets genomes and transcriptomes (*Figure 3C*). With connexins absent, lancelets could thus either have no gap junctions at all (see *Baatrup, 1981*; *Lane et al., 1987*; *Soledad Ruiz and Anadón, 1989*) or one formed from an innexin with an NGS. Both options, either the complete absence of regular gap junctions or the loss of all functional diversity among them are highly relevant for the evolution of

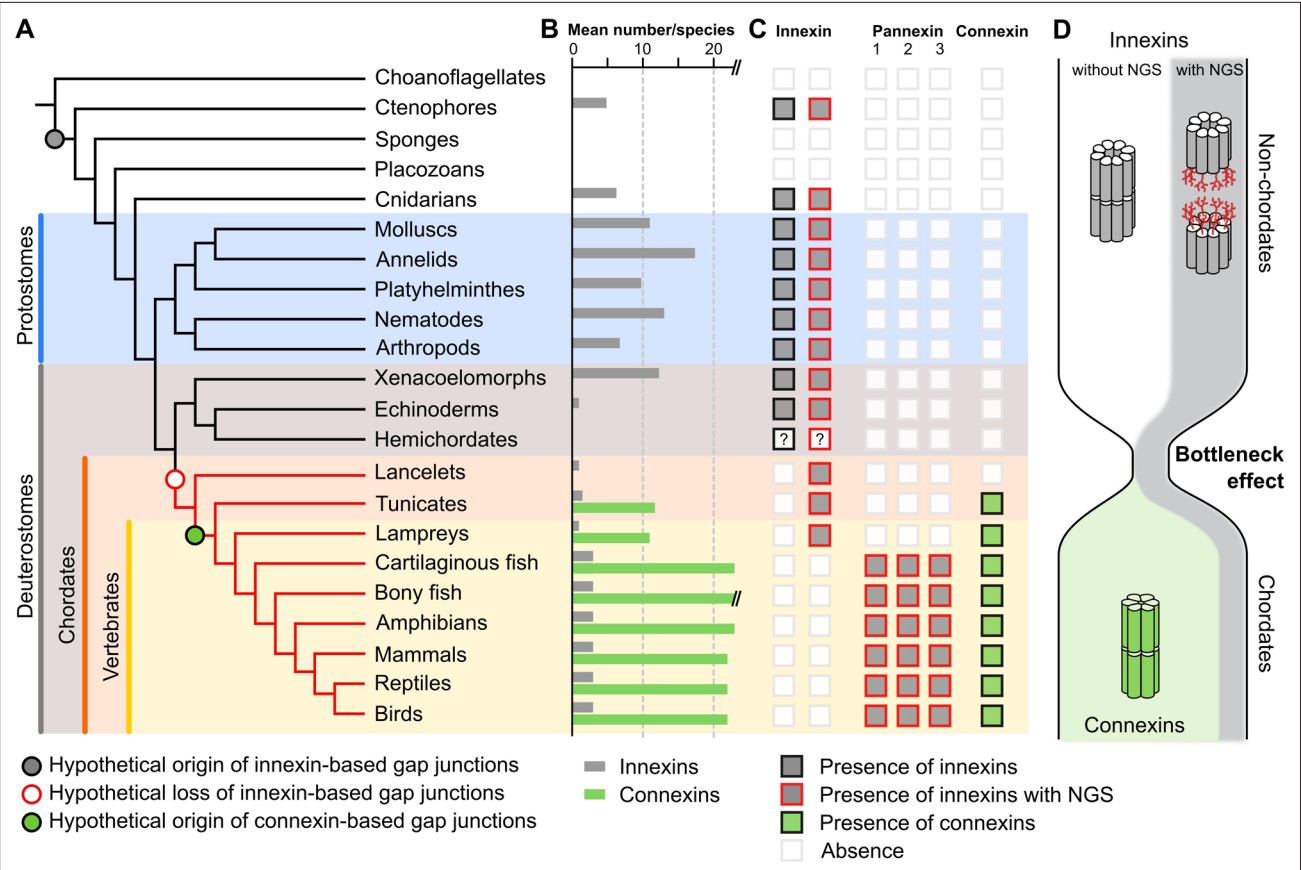

**Figure 3.** A new evolutionary scenario to explain the origin and exclusive use of connexin-based gap junctions in vertebrates. (**A**) Simplified phylogenetic tree to visualize the hypothetical loss of innexin-based gap junctions and the origin of connexin-based gap junctions. Lineages that only possess innexins with extracellular NGSs are indicated by a red branch color. (**B**) Mean number of different innexins and connexins per species in each of the different phyla. The Y-axis was cut for clarity (bony fish have 39–46 connexins due to genome duplications; *Mikalsen et al., 2021*). In hemichordates, we only found fragmented innexin-like sequences. (**C**) Occurrence of innexins and connexins in different metazoans. Innexins with an N-glycosylation consensus site (NGS) in their extracellular domains are present in chordates and non-chordates. Note that lancelets only possess one highly conserved innexin with one NGS and lack connexins. (**D**) A hypothetical evolutionary scenario in which the connexin-based gap junctions evolved after innexin diversity was lost during a bottleneck event in the early chordate evolution.

connexins. These arose de novo just at this time of chordate evolution (*Figure 3C*) and rapidly developed into diverse gene families (*Figure 3B*) with up to 22 connexins in mammals and 46 connexins in bony fish (*Mikalsen et al., 2021*) that again enabled a huge diversity of functionally unique gap junctions.

Based on our findings, we thus propose that a bottleneck effect at the origin of chordates might have been crucial for the evolution of the novel connexins (*Figure 3D*). In this evolutionary scenario, innexins were recruited as gap junction proteins in the last common ancestor of eumetazoa. While the innexins functionally diverge in ctenophores, cnidarians, and protostomes, the last common ancestor of the deuterostomes had lost all innexin diversity and thus the capacity to form functionally diverse gap junctions (*Figure 3C*). The strikingly high conservation of NGSs in lancelet, lamprey, and jawed vertebrate innexins (*Figure 2C–H*), their expression in every organ (*Penuela et al., 2014a*) and their association with a variety of diseases (*Esseltine and Laird, 2016*; *Penuela et al., 2014b*) suggest that the non-junctional innexin channels already served essential physiological functions in the last common ancestor of chordates and could not be converted into gap junctions. The loss of diversity in gap junctions is thus extreme, with just one, if any at all, functional gap junction. This finding and that of the strict conservation of the one remaining innexin could thus explain rather simply why the connexin family arose de novo and why it became the exclusive gap junction protein in all chordates although innexin-based gap junctions would have been fully capable to serve all functions (*Baker and Macagno, 2014*; *Bao et al., 2007*; *Bhattacharya et al., 2019*; *Lane et al., 2018*; *Liu et al.,*

*2016*; *Phelan and Starich, 2001*; *Skerrett and Williams, 2017*; *Welzel and Schuster, 2018*; *Yaksi and Wilson, 2010*) as they do so successfully in the sophisticated nervous systems of invertebrates (*Calabrese et al., 2016*; *Hall, 2017*; *Kristan et al., 2005*; *Marder et al., 2005*; *Otopalik et al., 2019*).

## Materials and methods
### Database searches
We used public databases to collect innexin amino acid sequences of chordate and non-chordate species. The taxonomic groups that we have analyzed in this study were constrained by the availability of publicly available genomic and transcriptomic data. We screened for innexin proteins across multiple taxa by using diverse sequences of the innexin family (PF00876) as sequence queries in BLAST searches. All retrieved sequences were further assessed and only innexin sequences that fulfilled all the following properties were included in our analyses: (1) the sequence was already assigned to the innexin family (PF00876) or a reciprocal BLAST with the sequence hit as query against the UniProt database identified a known innexin sequence as a top hit; (2) the sequence is predicted to contain four transmembrane domains that are connected by two extracellular and one intracellular loop as well as an intracellular N- and C-terminus (see *Figure 1A*). To clarify this, we used the TMHMM Server v2.0 (https://services.healthtech.dtu.dk/service.php?TMHMM-2.0) to predict membrane topology; (3) the sequence is not fragmented or a duplicate entry. In total, we retrieved 1521 innexin protein sequences of nine non-chordate groups (phylum ctenophores, phylum cnidarians, phylum molluscs, phylum annelids, phylum plathyhelminthes, phylum nematodes, phylum arthropods, phylum xenacoelomorphs, and phylum echinoderms) and 867 sequences of nine chordate groups (subphylum lancelets, subphylum tunicates, class lampreys, class cartilaginous fish, superclass bony fish, class amphibians, class reptiles, class birds, and class mammals). All innexin sequences of molluscs, annelids, plathyhelminthes, nematodes, arthropods, cartilaginous fish, bony fish, amphibians, reptiles, birds, and mammals were obtained from the protein databases at NCBI (http://www.ncbi.nlm.nih.gov) and UniProt (http://www.uniprot.org). The innexin sequences of the ctenophore species were obtained from the Neurobase genome database (http://neurobase.rc.ufl.edu/Pleurobrachia). The innexin sequences of the cnidarian species were obtained from UniProt and the Marimba genome database (http://marimba.obs-vlfr.fr). The LanceletDB database (http://genome.bucm.edu.cn/lancelet) was used to retrieve innexin sequences of lancelets. The innexin sequences of the tunicate species were obtained from NCBI and the ANISEED database (https://www.aniseed.cnrs.fr). The innexin sequences of lampreys were retrieved from the NCBI and the SIMRBASE database (https://genomes.stowers.org). We additionally retrieved ctenophore, cnidarian, xenacoelomorph, and echinoderm innexin sequences from the NCBI TSA database. The full list of species and taxa, along with accession numbers and links to the corresponding databases can be found in *Figure 1—source data 1* and *Figure 2—source data 1*.

We searched for connexins in all non-chordate taxa as well as in the non-vertebrate chordates by using BLAST searches against NCBI Transcriptome Shotgun Assembly (TSA) databases and NCBI genome databases. None of the non-chordate species showed connexin-like proteins. The only case with an apparently connexin-like sequence in the transcriptome was in the red sea urchin (*M. franciscanus*; GHJZ01010392.1). However, we could neither confirm this sequence in the red sea urchin genome nor did the analysis of the transcriptome of 41 other echinoderms, including two closely related species, ever reveal any connexin-like sequence. We therefore have decided not to include the transcriptome-based sequence of the red sea urchin into our analysis.

### Identification of potential N-glycosylation sites
To identify potential N-glycosylation sites within the ELs of the non-chordate and chordate innexins, we generated 16 multiple sequence alignments for each taxonomic group (7 non-chordate and 9 chordate groups). For each group, we first imported all innexin protein sequences of each species into the Jalview software (version 2.11.1.4) (*Waterhouse et al., 2009*). The innexin sequences obtained from the UniProt database were automatically retrieved into Jalview by the UniProt sequence fetcher. The sequences obtained from other databases were manually added to Jalview. After aligning the innexin sequences with ClustalW (*Thompson et al., 1994*), the resulting multiple sequence alignments of each group were used to identify potential N-glycosylation consensus

sites (NGS) in the extracellular domains of each innexin protein. NGSs in innexins were predicted by the NetNGlyc 1.0 Server (https://services.healthtech.dtu.dk/service.php?NetNGlyc-1.0) that uses an artificial neural network to examine the sequence context of the N-X-S/T motif. We used the following criteria to include the NGS into our analyses: (1) X in the N-X-S/T motif could be any amino acid except proline; (2) the potential score was >0.5 and the agreement between the nine artificial neural networks was ≥5/9; and (3) the NGS was located in EL1 or EL2. The positions of all extracellular N-glycosylation sites are reported in *Figure 1—source data 1* and *Figure 2—source data 1*.

### Phylogenetic tree construction

We visualized the incidence of innexins with N-glycosylation motif in their extracellular domains within different taxonomic groups by using phylogenetic trees. To generate a phylogenetic tree of the 1468 non-chordate and the 867 chordate innexins, respectively, we first created two global alignments including the available alignments of the seven non-chordate or the nine chordate groups. Both alignments were generated using MEGA version X (*Kumar et al., 2018*; *Stecher et al., 2020*) with the default parameters of ClustalW. Both multiple sequence alignments were then processed by the G-blocks server (http://molevol.cmima.csic.es/castresana/Gblocks.html) (*Castresana, 2000*) to automatically detect and remove poorly aligned, nonhomologous, and excessively divergent alignment columns. We reconstructed a phylogenetic tree of the non-chordate and the chordate innexins, respectively, by using the raxmlGUI 2.0 software (*Edler et al., 2020*). Before the phylogenetic analyses, ModelTest-NG (*Darriba et al., 2020*) was run on the two trimmed alignments with the default parameters to determine the best probabilistic model of sequence evolution. Both phylogenetic trees were built using the maximum likelihood method based on the JTT model and 100 bootstrap replications. The phylogenetic trees of chordate and non-chordate innexins were visualized, edited, and annotated with iTOL v5 (https://itol.embl.de) (*Letunic and Bork, 2021*).

## Acknowledgements

The authors thank Antje Halwas for generating the multiple sequence alignments, David Richter for expert advice on genome and transcriptome analysis, and Andreas Möglich for inspiring discussions.

## Additional information

### Funding

| Funder | Grant reference number | Author |
| --- | --- | --- |
| Deutsche Forschungsgemeinschaft | Schu1470/8 491183248 | Stefan Schuster |
| University of Bayreuth | Open Access Publishing Fund | Georg Welzel Stefan Schuster |

The funders had no role in study design, data collection and interpretation, or the decision to submit the work for publication.

### Author contributions

Georg Welzel, Conceptualization, Formal analysis, Investigation, Methodology, Visualization, Writing - original draft; Stefan Schuster, Conceptualization, Resources, Supervision, Validation, Writing - review and editing

### Author ORCIDs

Georg Welzel [ID] http://orcid.org/0000-0002-7017-604X
Stefan Schuster [ID] http://orcid.org/0000-0002-0873-8996

### Decision letter and Author response
Decision letter https://doi.org/10.7554/eLife.74422.sa1
Author response https://doi.org/10.7554/eLife.74422.sa2

## Additional files

### Supplementary files
• Transparent reporting form

### Data availability
All data generated or analysed during this study are included in the manuscript and in the supporting files. Source Data files have been provided for Figure 1 and 2.

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
