## [Decision Letter]

**Decision letter after peer review:**

Thank you for submitting your article "Connexins evolved after early chordates lost innexin diversity" for consideration by *eLife*. Your article has been reviewed by 3 peer reviewers, and the evaluation has been overseen by a Reviewing Editor and Patricia Wittkopp as the Senior Editor. The following individual involved in the review of your submission as has agreed to reveal their identity: Hisayo Sadamoto (Reviewer #3)

Essential revisions:

1) Please re-analyse in more detail genomes and also available transcriptomes from xenambulacrarians. Understanding the distribution of connexins and innexins in this clade is particularly important for accurately reconstruction the evolutionary steps. This is especially important, in light of the identification of a connexin in a sea urchin by reviewer #3.

2) Discuss the data also in light of EM evidence for gap junctions in chordates and ambulacrarians.

3) Given that there is no experimental evidence that the glycosylated innexins of Lancelets and Tunicates prevents gap junction formation, glycosylated innexins may still form gap junctions in these organisms. Please critically discuss this possibility and its implications for the scenario.

*Reviewer #1 (Recommendations for the authors):*

I have no suggestions for additional experiments/analysis.

Recommendations for improving the writing and presentation:

1. Figure 2: The conservation criteria are not completely clear: column 7 of absolutely conserved G is labelled with + and not with *, while column 5 of K/R residues is not labelled with a +.

2. Line 201: «lancelets, which do not yet express connexins» – "yet" implies that at some point lancelets will start expressing the connexins which is not true.

3. Line 217: "non-junctional innexin channels already served essential physiological functions in the basal chordates» – using terms «last common ancestor of chordates» or «last common ancestor of vertebrates» would be more precise. I assume that Lancelets and Tunicates are referred to as "basal chordates" here however they are currently living animals and are not ancestors of vertebrates.

*Reviewer #2 (Recommendations for the authors):*

"Animals from hydra to human"

If ctenophores have innexins and gap junctions, this statement excludes them – change to from humans to comb jellies

The authors write that:

"However, some of the innexins with identified NGSs have previously been shown to form functional gap junction channels (Figure 1F and Supplementary File 3). This means that either the predicted NGSs of these innexins are not glycosylated (Apweiler, 1999) or that glycosylation does not necessarily entail the loss of gap junction function in the diverse innexins of invertebrates."

This slightly contradicts their statement that NGS prevents the formation of gap junctions. Can the authors analyse and discuss which sites are glycosylated in these cases and if there is a difference between the site of NG and the influence on gap junction formation.

"Moreover, we show that in vertebrates the sequence of the extracellular loops as well as the positions of the glycosylation motifs are highly conserved (Figure 2C-F)."

Please discuss the situation in non-vertebrate chordates.

"in the common cnidarian/bilaterian ancestor" – what about ctenophores?

Language:

"during the early chordate evolution" during the early evolution of chordates or during early chordate evolution.

"early in the metazoan evolution" early in metazoan evolution.

*Reviewer #3 (Recommendations for the authors):*

1) For animal phyla for which accurate sequence data are scarce, an additional search that includes TSA will yield better results. The absence of both innexin and connexin genes in Echinoderms were reported (Figure 3, Slivko-Koltchik et al., 2019), however, it is unlikely that no gap junction proteins present at all. Indeed. I performed a blast search of echinoderm TSA data using mammalian connexin sequences, and found one connexin (connexin 30)-like sequence in red sea urchin in 2019 (https://www.ncbi.nlm.nih.gov/nuccore/ GHJZ01010392.1/). The addition of this echinoderm data to the present result, will change the result of Figure 3, which showed a clear switching point between innexin and connexin. It suggests the existence of an ambiguous evolutionary process with only one of innexin or connexin, which requires a new discussion.

2) The authors proposed a scenario in which connexins arosed due to the loss of gap junction forming ability of innexins during the early chordate evolution. However, there is no consideration of the number of innexin and connexin family proteins per each animal species.

Gap junction channels are composed of two hemichannels, which are formed from six to eight connexin/innexin molecules. In many invertebrate species, there are at least five to ten different types of innexin family proteins per species. In vertebrates, each animal species has more than 10 connexin family proteins. It is considered that the combination of these connexin/innexin family proteins finely regulates the channel function at gap junctions. In Echinoderms and Lancelets, both of innexin and connexin decreased the number and variety of family proteins, however, it cannot be explained by a bottleneck effect of innexin. More, the loss of variety of gap junction proteins means that it becomes poor as a channel regulatory mechanism, even though Echinoderms and Lancelets are highly evolved and can be expected to have advanced regulatory mechanisms as multicellular organisms. That seems more reasonable to assume that there is a third gap junction protein except innexin and connexin. I would like to read the discussion about this.

3) The following paper recently reported the classification of innexin sequences and the glycosylation modifications on them in the terrestrial slug (*Limax valentianus*). The methods are almost the same as in this paper. The gene transcripts of mollusc innexin homologs were characterized based on sequence alignment, phylogenetic analysis, and protein modification site prediction. I recommend that the authors add this paper as a reference to Lines 58-59.

Identification and classification of innexin gene transcripts in the central nervous system of the terrestrial slug *Limax valentianus*

Sadamoto H, Takahashi H, Kobayashi S, Kondoh H, Tokumaru H (2021) Identification and classification of innexin gene transcripts in the central nervous system of the terrestrial slug Limax valentianus. PLOS ONE 16(4): e0244902. https://doi.org/10.1371/journal.pone.0244902

4) The figure legend (line 168-169, Figure 2A) "Black and red branches" should be "Red branches".

---

## [Author Response]

Essential revisions:1) Please re-analyse in more detail genomes and also available transcriptomes from xenambulacrarians. Understanding the distribution of connexins and innexins in this clade is particularly important for accurately reconstruction the evolutionary steps. This is especially important, in light of the identification of a connexin in a sea urchin by reviewer #3.

We fully agree and were also curious about the situation in the xenambulacrarians. Thanks to the excellent suggestions of the reviewers, we were able to extend the analysis by adding 118 innexins from 24 additional species: 3 ctenophores, 11 cnidarians, 4 xenacoelomorphs, 4 echinoderms and 2 lancelets. Perhaps most notably, we did identify innexins in the xenambulacrarians, thus solving the long-standing puzzle how they manage to form gap junctions. In the sea stars and the sea-urchins none of the 41 species we analyzed, including very close relatives of the red sea urchin, had connexins. In the red sea urchin, we also identified a connexin-like sequence in the transcriptome database but not in the genome database. At any rate, this would be an extremely interesting but also completely isolated case of a connexin in an invertebrate.

2) Discuss the data also in light of EM evidence for gap junctions in chordates and ambulacrarians.

The major surprise here was that we were able to identify, apparently for the first time, innexins in ambulacrarian transcriptomes. This should also end the long discussion of how ambulacrarians form gap junctions (as seen in EM) in the apparent absence of either innexins or connexins.

3) Given that there is no experimental evidence that the glycosylated innexins of Lancelets and Tunicates prevents gap junction formation, glycosylated innexins may still form gap junctions in these organisms. Please critically discuss this possibility and its implications for the scenario.

Thank you very much for this remark! Yes, we agree and fully allow for this possibility. After all, our point is that the lancelets, with only one glycosylated innexin remaining, completely lose all functional diversity in any gap junctions they might have (if any at all). With the glycosylated innexins so highly conserved, evolving diversity of functions thus requires new gap junction proteins. We hope that this point is now made much clearer (with a variety of changes, starting in the introduction).

Reviewer #1 (Recommendations for the authors):I have no suggestions for additional experiments/analysis.Recommendations for improving the writing and presentation:1. Figure 2: The conservation criteria are not completely clear: column 7 of absolutely conserved G is labelled with + and not with *, while column 5 of K/R residues is not labelled with a +.

Thank you for pointing this out. The conservation criteria refer to the complete alignments of all 867 innexins and not only to the representative sequences shown in Figure 2. We have clarified this point in in the legend of Figure 2.

2. Line 201: «lancelets, which do not yet express connexins» – "yet" implies that at some point lancelets will start expressing the connexins which is not true.

Yes, we agree (we have removed “yet”).

3. Line 217: "non-junctional innexin channels already served essential physiological functions in the basal chordates» – using terms «last common ancestor of chordates» or «last common ancestor of vertebrates» would be more precise. I assume that Lancelets and Tunicates are referred to as "basal chordates" here however they are currently living animals and are not ancestors of vertebrates.

We agree and have revised the statement (page 9, line 211).

Reviewer #2 (Recommendations for the authors):"Animals from hydra to human"If ctenophores have innexins and gap junctions, this statement excludes them – change to from humans to comb jellies

We have changed the statement in the revised manuscript (page 2, line 27).

The authors write that:"However, some of the innexins with identified NGSs have previously been shown to form functional gap junction channels (Figure 1F and Supplementary File 3). This means that either the predicted NGSs of these innexins are not glycosylated (Apweiler, 1999)or that glycosylation does not necessarily entail the loss of gap junction function in the diverse innexins of invertebrates."This slightly contradicts their statement that NGS prevents the formation of gap junctions. Can the authors analyse and discuss which sites are glycosylated in these cases and if there is a difference between the site of NG and the influence on gap junction formation.

We have re-analyzed the NGS of the innexins with gap junction function and discussed our findings in more detail in the revised manuscript (page 5, line 123) and included more information in Figure 1- source data 3.

"Moreover, we show that in vertebrates the sequence of the extracellular loops as well as the positions of the glycosylation motifs are highly conserved (Figure 2C-F)."Please discuss the situation in non-vertebrate chordates.

As mentioned above, we have included innexin alignments of non-vertebrate chordates in Figure 2 and discuss these in the revised manuscript.

"in the common cnidarian/bilaterian ancestor" – what about ctenophores?

We have changed the statement in “the last common ancestor of eumetazoa”

Language:"during the early chordate evolution" during the early evolution of chordates or during early chordate evolution."early in the metazoan evolution" early in metazoan evolution.

Thank you for pointing this out. We have changed it in the revised manuscript (page 8, line 204)

Reviewer #3 (Recommendations for the authors):1) For animal phyla for which accurate sequence data are scarce, an additional search that includes TSA will yield better results. The absence of both innexin and connexin genes in Echinoderms were reported (Figure 3, Slivko-Koltchik et al., 2019), however, it is unlikely that no gap junction proteins present at all. Indeed. I performed a blast search of echinoderm TSA data using mammalian connexin sequences, and found one connexin (connexin 30)-like sequence in red sea urchin in 2019 (https://www.ncbi.nlm.nih.gov/nuccore/ GHJZ01010392.1/). The addition of this echinoderm data to the present result, will change the result of Figure 3, which showed a clear switching point between innexin and connexin. It suggests the existence of an ambiguous evolutionary process with only one of innexin or connexin, which requires a new discussion.

Thank you for your valuable hints! We searched for innexins in echinoderms by using the TSA databases and found an innexin in *Asterias rubens* as well as in three other echinoderms. Thus, echinoderms obviously use innexin-based gap-junctions which can now also explain the electrical coupling of cells in *Asterias rubens* described by Slivko-Koltchik et al., 2019.

We also found the connexin-like sequence in the red sea urchin. However, we neither found this sequence in the red sea urchin genome nor in the transcriptome of 41 other echinoderms. Based on these data, it is unlikely that echinoderms have connexins and we therefore decided to not include the single connexin-like sequence in the results of Figure 3 but directly discuss this in line 102, page 5.

2) The authors proposed a scenario in which connexins arosed due to the loss of gap junction forming ability of innexins during the early chordate evolution. However, there is no consideration of the number of innexin and connexin family proteins per each animal species.Gap junction channels are composed of two hemichannels, which are formed from six to eight connexin/innexin molecules. In many invertebrate species, there are at least five to ten different types of innexin family proteins per species. In vertebrates, each animal species has more than 10 connexin family proteins. It is considered that the combination of these connexin/innexin family proteins finely regulates the channel function at gap junctions. In Echinoderms and Lancelets, both of innexin and connexin decreased the number and variety of family proteins, however, it cannot be explained by a bottleneck effect of innexin. More, the loss of variety of gap junction proteins means that it becomes poor as a channel regulatory mechanism, even though Echinoderms and Lancelets are highly evolved and can be expected to have advanced regulatory mechanisms as multicellular organisms. That seems more reasonable to assume that there is a third gap junction protein except innexin and connexin. I would like to read the discussion about this.

Thank you for pointing this out. As already mentioned above, we have discussed the potential loss of heteromeric/heterotypic gap junction channels in the revised manuscript.

As described above, we actually found innexins in echinoderm transcriptomes, so that innexin-based gap junctions would be possible within this group.

3) The following paper recently reported the classification of innexin sequences and the glycosylation modifications on them in the terrestrial slug (Limax valentianus). The methods are almost the same as in this paper. The gene transcripts of mollusc innexin homologs were characterized based on sequence alignment, phylogenetic analysis, and protein modification site prediction. I recommend that the authors add this paper as a reference to Lines 58-59.Identification and classification of innexin gene transcripts in the central nervous system of the terrestrial slug Limax valentianusSadamoto H, Takahashi H, Kobayashi S, Kondoh H, Tokumaru H (2021) Identification and classification of innexin gene transcripts in the central nervous system of the terrestrial slug Limax valentianus. PLOS ONE 16(4): e0244902. https://doi.org/10.1371/journal.pone.0244902

We have added this paper to the revised manuscript (page 3, line 65).

4) The figure legend (line 168-169, Figure 2A) "Black and red branches" should be "Red branches".

You are right, we have corrected the legend of Figure 2A in the revised manuscript.